# Exploring a Rarity: Incidence of and Therapeutic Approaches for Neurological Complications and Hypophysitis in Cancer Patients on Immune Checkpoint Inhibitors—A Single-Center Study

Anna Lea Amylidi [1,2,*], Aristeidis Gogadis [1,2], Melina Yerolatsite [1,2], George Zarkavelis [1,2], Nanteznta Torounidou [1,2], Varvara Keramisanou [1,2], Eleftherios Kampletsas [1,2] and Davide Mauri [1,2,3]

1   Department of Medical Oncology, University Hospital of Ioannina, 45500 Ioannina, Greece; agogadis@gmail.com (A.G.); m.yerolatsite@gmail.com (M.Y.); gzarkavelis@outlook.com (G.Z.); nadia.torou@gmail.com (N.T.); barbiekermds@gmail.com (V.K.); lekable4@yahoo.gr (E.K.); dvd.mauri@gmail.com (D.M.)
2   Society for Study of Clonal Heterogeneity of Neoplasia (EMEKEN), 45445 Ioannina, Greece
3   Faculty of Medicine, School of Health Sciences, University of Ioannina, 45500 Ioannina, Greece
*   Correspondence: annalea.ami@gmail.com; Tel.: +30-694477774

**Abstract:** Immune checkpoint inhibitors, such as anti-PD-1 and anti-CTLA-4 inhibitors, have become the standard of care for many cancer types. However, they induce immune-related adverse events (irAEs), including neurotoxicity and hypophysitis. The incidence and outcomes of neurotoxicity and hypophysitis in patients treated with immune checkpoint inhibitors are not well established. We conducted a retrospective study of 812 patients with solid cancers who received immune checkpoint inhibitors at the University General Hospital of Ioannina between January 2018 and January 2023. We assessed demographic and clinical data, including the severity of symptoms, treatment regimen, other irAEs, resolution type and time, and death. Two patients experienced neurotoxicity and two hypophysitis. All four patients required inpatient administration and received corticosteroids or/and hormone replacement. Three patients responded to the initial therapy, experiencing full recovery, while one patient was corticosteroid-resistant, and immunoglobin G was administered. Two patients never received immunotherapy after their toxicity due to the severity of symptoms; one patient continued monotherapy with nivolumab, changing from combination therapy with ipilimumab–nivolumab, while the fourth patient continued his initial treatment with nivolumab. Our study suggests that the incidence of neurotoxicity and hypophysitis in patients treated with immune checkpoint inhibitors is low, but careful monitoring and prompt treatment with corticosteroids are necessary for effective management.

**Keywords:** hypophysitis; neurotoxicity; checkpoint inhibitor; aseptic meningitis; encephalitis; immune-related adverse events

## 1. Introduction

In the last few decades, the introduction of immune checkpoint inhibitors (ICIs) transformed the cancer treatment landscape [1–3]. The foundation of immunotherapy lies in its capacity to identify abnormal tissue and bolster the body's immune system against tumor cells. The immune system features both stimulatory and inhibitory elements governing the generation of immune responses, maintaining a delicate balance to prevent auto-immune reactions towards self-antigens through the positive selection of T cells. However, tumor cells can exploit this distinctive mechanism by modulating the activity of tumor environment cells, particularly T-cells, either through inhibition or hyperstimulation. Unlike other forms of anticancer therapies, immune checkpoint inhibitors specifically target both stimulatory and inhibitory T-cell receptors, resulting in T-cell activation and eliciting antitumor responses [4,5]. Various immune checkpoints involved

in cell proliferation, such as CTLA-4 (cytotoxic T-lymphocyte-associated antigen 4), PD-1/PDL-1 (programmed cell death 1/programmed cell death ligand pathway), TIM-3 (T-cell immunoglobulin- and mucin-domain-containing molecule 3), TIGIT (T-cell immunoreceptor with immunoglobulin and immunoreceptor tyrosine-based inhibitory motif domain), and LAG-3 (lymphocyte-activation gene-3), play crucial roles in this intricate regulatory network [6,7].

ICIs are monoclonal antibodies that target the immune checkpoint receptors. The first FDA-approved immune checkpoint inhibitor was ipilimumab (anti-CTLA-4), which was used to treat melanoma [8]. Following the introduction of CTLA-4 inhibitors, subsequent approvals were granted for anti-PD-1 (e.g., pembrolizumab, nivolumab, camrelizumab), anti-PDL-1 (e.g., atezolizumab, avelumab), and most recently, for anti-LAG-3 (e.g., relatimab) inhibitors, whether used alone or in combination [6,9,10]. This progression underscores the pivotal role of ICIs as the cornerstone of contemporary anticancer therapy.

However, ICIs can induce several immune-related adverse events that can potentially affect every single organ as a consequence of overactivation of the immune system and T-cells [11,12]. The exact mechanism remains partially unclear [13,14]; immune-related adverse events frequently include endocrine, cutaneous, and gastrointestinal toxicities [15–18]. Hepatotoxicity, pulmonary toxicity, rheumatologic toxicity, cardiovascular complications, kidney injury, and ocular toxicities as well as hematologic toxicities have been documented with the use of immune checkpoint inhibitors [15,18,19]. On some occasions and according to the tissue or organ affected and the extent of damage, these adverse events may be fatal [20]. The median onset time of irAE presentation varies between 4 and 14 weeks depending on the type of irAE and the ICI regimen that is used [21–24].

Serious adverse effects on the central and peripheral nervous system are rare but potentially lethal and require prompt recognition and treatment from clinicians [20]. Adverse events related to the nervous system are estimated to occur in approximately 1–5% of cases, while the incidence of hypophysitis associated with immune checkpoint inhibitors is estimated at 1–10% [20–23]. A diverse range of neurological adverse events has been documented, affecting both the central and peripheral nervous systems. Nearly half of these events fall under neuromuscular disorders, encompassing conditions such as myositis, myasthenia gravis, demyelinating disorders, and overlapping expressions of these entities. The predominant central nervous system disorders linked to immunotherapy include encephalitis, vasculitis, aseptic meningitis, transverse myelitis, cranial neuropathies, and various demyelinating syndromes [24,25].

The exact mechanism of central nervous system (CNS) deficits following immunotherapy administrations is not quite known. PD-1 and CTLA-4 cell expression and tumor micro-environment accumulation vary. According to published data, the expression of both receptors has been documented in the pituitary gland [26]. The expression of these molecules on T-regulatory cells and the corollary loss of immune regulation upon immunotherapy administration is speculated to play a significant role in CNS immune-related adverse events [27]. Pro-inflammatory and inflammatory cytokine levels in the CNS have also been implied to participate in the onset and evolution of neurological immune adverse events [28]. In cases of hypophysitis, a two-step hypersensitivity reaction has been described, consisting of an early complement cascade activation and a further penetration of the gland by autoreactive lymphocytes [29]. Another proposed mechanism that has been described includes the cross-reactivity phenomenon, owing to similarities in tumor antigen and nervous system cell epitopes. Autoimmunity exacerbation, either due to epitope spreading or specific antibody proliferation, is also being investigated as a trigger of adverse events [30,31]. Patients' genetic factors have also been hypothesized to affect the impact of immunotherapy on the nervous system [32]. However, it is of note that clinical trials leading to immunotherapy approvals did not include patients with underlying autoimmune diseases and the rapidly expanding application of immunotherapy will ultimately lead to accumulating more data that will shed light upon the background of immune-related toxicity.

Hypophysitis typically presents 6–8 weeks after the initiation of immunotherapy, with a wide range of unspecific symptoms such as headaches, fatigue, nausea, weight loss, generalized muscle weakness, and mental state changes [20,27,33]. It is more frequently observed in patients receiving anti-CTLA-4 therapy than in patients receiving anti-PDL-1/PD-1 therapy [12]. An MRI of hypophysis, as well as a full endocrine blood workup, should be carried out to exclude other possible causes and to establish the diagnosis. Once the diagnosis is made, hormone replacement is enough, and the treatment can be safely continued [34]. Furthermore, aseptic meningitis can occur in 0.36% of patients who are on ICI regimens [21–24]. It often presents with neck stiffness, fever, and headache, but without alteration of consciousness. All patients need to undergo a brain MRI, cerebral spine fluid (CSF) analysis, and full blood workup to rule out other possible pathologies and to immediately start intravenous corticosteroids, immunoglobulins, or/and other immunosuppression drugs. Whilst encephalitis symptoms resemble aseptic meningitis, patients need to also have altered mental status symptoms, such as confusion, seizures, and/or cerebellar ataxia. The outcome, if not treated immediately, can be fatal. Differential diagnosis is similar to that of aseptic meningitis, as is the treatment [15].

This study aims to ascertain the occurrence rate of rare CNS adverse events and hypophysitis among individuals with solid cancers undergoing anti-PD1 and/or anti-CTLA-4 treatment at our institution. Additionally, we outline the treatment approach implemented at our university hospital, examine the mortality associated with these irAEs, and analyze survival outcomes following a severe irAE.

## 2. Materials and Methods

In this retrospective analysis, we systematically reviewed the medical records of individuals with solid cancers who underwent treatment at the University General Hospital of Ioannina with either anti-PD1 inhibitors or anti-CTLA-4 (ipilimumab) inhibitors. This retrospective study obtained an ethics approval from the Institutional Review Board of the University Hospital of Ioannina. The study period spanned from January 2018 to January 2023. A comprehensive evaluation of the medical records was conducted by three independent investigators (A.L.A., A.G., M.Y.). Only patients with a diagnosis of central nervous system (CNS) immune-related adverse events (irAEs) or autoimmune hypophysitis of any grade were included in the analysis. The irAEs were graded according to the Common Terminology Criteria for Adverse Events (CTCAE) version 5.0. [35] Furthermore, demographic information, severity of symptoms, details regarding the treatment regimen administered, concurrent irAEs, resolution patterns and timelines of symptoms, as well as instances of mortality from any cause were meticulously documented in an anonymized manner for each patient.

## 3. Results

Between January 2018 and January 2023, a total of 812 patients were treated with immunotherapy at our hospital. We included individuals who received a minimum of one infusion of either ipilimumab, nivolumab, pembrolizumab, or a combination of ipilimumab–nivolumab. A total of 463 patients (57%) received pembrolizumab monotherapy or in combination with a tyrosine kinase inhibitor (3.2%), 281 (34.7%) received nivolumab monotherapy or in combination with cabozantinib (4.2%), 67 (8.3%) received ipilimumab plus nivolumab, and 1 patient received ipilimumab. In terms of sex distribution, 80.9% of the patients were male and 19.1% were female. The prevailing malignancies included non-small-cell lung cancer in 442 patients (54.4%), renal cancer in 81 patients (9.9%), melanoma in 63 patients (7.7%), and bladder cancer in 62 patients (7.6%) (Table 1).

**Table 1.** Characteristics of patients who received immunotherapy at our institute.

| | *N* = 812 (%) |
|---|---|
| Sex | |
| Female | 155 (19.1%) |
| Male | 657 (80.9%) |
| Type of Treatment | |
| Pembrolizumab | 463 (57%) |
| Combination pembrolizumab and TKI | 3.2% 25 |
| Nivolumab | 281 (34.7%) |
| Combination nivolumab and cabozatinib | 4.2% 34 |
| Combination nivolumab and ipilimumab | 67 (8.3%) |
| Ipilimumab | 1 (0.12%) |
| Type of malignancies | |
| Non-small-cell lung cancer | 442 (54.4%) |
| Renal cancer | 81 (9.9%) |
| Melanoma | 63 (7.7%) |
| Bladder cancer | 62 (7.6%) |

In total, out of 812 patients, only 2 exhibited ICI-related CNS adverse events, and an additional 2 patients experienced hypophysitis. This corresponds to a mere 0.25% incidence among our ICI-treated patients for CNS irAEs and 0.25% for hypophysitis as irAE (2/812). The median age of the affected individuals was 57 years, with an equal distribution between the sexes. Among these patients, two received a combination of nivolumab plus ipilimumab, while the other two were treated with pembrolizumab. Three individuals experienced serious adverse events related to immune checkpoint inhibitor (ICI) treatment, occurring more than three months into the ICI therapy. Meanwhile, only one patient exhibited a swift onset of symptoms following the initial ICI infusion. Detailed patient descriptions are reported separately below (Table 2).

**Table 2.** Patient characteristics. AE: adverse event.

| | Age | Type of Tumor | Type of Immunotherapy | Time Onset AE | Final Diagnosis | Treatment of AE | Recovery | Discontinuation Of Immunotherapy | Outcome | AE Grading |
|---|---|---|---|---|---|---|---|---|---|---|
| Case 1 | 43 y | Metastatic melanoma | Anti-CTLA 4+ Anti-PD1 | 2 weeks | Meningoencephalitis + hepatitis | Methylprednisolone 2 mg/kg/day | Fully recovered | Yes | Died after 12 months | 3 |
| Case 2 | 56 y | Metastatic NSCLC | Anti-PD1 | 21 weeks | Aseptic meningitis | Methylprednisolone 2 mg/kg/day and immunoglobin G 2 g/kg | Partially recovered | Yes | Died after 18 months | 4 |
| Case 3 | 61 y | Metastatic NSCLC | Anti-PD1 | 39 weeks | Hypophysitis | Hydrocortisone and hormone replacement | Fully recovered | No | Alive | 3 |
| Case 4 | 68 y | Pleural mesothelioma | Anti-CTLA 4+ Anti-PD1 | 13 weeks | Hypophysitis | Hydrocortisone | Fully recovered | No | Alive | 3 |

*3.1. Case 1*

A 43-year-old female was diagnosed in 2015 with a subungual lesion in the right index finger. Subsequent to surgical excision, a histopathology report confirmed the presence of subungual melanoma, leading to the administration of interferon A as adjuvant therapy. However, after 4 years, two lesions, one in the lung and one in the right arm, were excised and confirmed as melanoma recurrence (BRAF wild type). Consequently, the patient commenced immunotherapy with nivolumab in July 2019. In April 2020, due to hepatic disease relapse, hepatectomy of sections V and VI was performed. Post-hepatectomy staging

revealed periportal hepatic lymphadenopathy, prompting the initiation of combination immunotherapy with nivolumab (1 mg/kg) and ipilimumab (3 mg/kg) in June 2020. Two weeks after the first infusion, the patient presented to the ER with high fever, disorientation, and neck stiffness. Laboratory analysis indicated significantly elevated liver enzymes (AST, ALT, 20× ULN) and CRP. Notably, viral testing (HBV, HCV, HIV, EBV, CMV, HSV2) and thyroid function yielded normal results. Brain CT and liver ultrasound exhibited no pathological findings. Empirical administration of ceftriaxone and acyclovir ensued promptly. Cerebral spinal fluid (CSF) examination revealed elevated total protein (281 mg/dL), normal glucose, and 148 cells per mm$^3$, with lymphocyte predominance (84%). CSF cultures, viral examination, and cytology yielded negative results (Table 3). Based on the clinical presentation, radiological and laboratory findings, and neurology input, the patient received a diagnosis of meningoencephalitis (Grade 3-CTCAE v 5.0) and autoimmune hepatitis as adverse events of combination immunotherapy. Methylprednisolone was initiated at 2 mg/kg/day, with no improvement until the fourth day post administration. During the 15-day hospitalization, the patient fully recovered from neurologic symptoms, and liver enzymes normalized. Subsequently, she did not undergo further immunotherapy and succumbed one year later due to liver and brain metastasis.

**Table 3.** Summaries of case presentations.

| | Symptoms | CFS Examination | CFS Culture | Viral Examination | CFS Cytology | Paraneoplastic Antibodies | Laboratory Testing | Electromyogram | MRI Findings |
|---|---|---|---|---|---|---|---|---|---|
| Case 1 | Fever, disorientation, and neck stiffness | TP 281 mg/dL Glu 49 mg/dL 148 cells/mm$^3$ with lymphocyte predominance | Negative | Negative | Negative | NA | NA | NA | No pathological findings |
| Case 2 | Dysarthria and disequilibrium | TP 112 mg/dL Glu 87 mg/dL 20 cells/mm$^3$ with lymphocyte predominance | Negative | Negative | Negative | Negative | NA | NA | Micro-ischemic-type lesions |
| Case 3 | Muscle weakness in both lower limbs | NA | NA | NA | NA | NA | Low serum cortisol levels, low ACTH, low TSH | Negative | Enlargement of the pituitary gland and progressively enhancing bilateral ring-shaped enhancement |
| Case 4 | Severe exhaustion, anorexia, dizziness, and nausea | NA | NA | NA | NA | NA | Low serum cortisol levels, low ACTH, low TSH | NA | Mild swelling of the pituitary gland |

CSF: Cerebrospinal fluid, Glu: Glucose, MRI: Magnetic resonance imaging, NA: Not applicable, TP: Total protein.

### 3.2. Case 2

A 56-year-old man was diagnosed with stage IV non-small-cell lung cancer (NSCLC), PDL 1 < 1%, and in January 2020 started chemotherapy combined with pembrolizumab. Twenty-one weeks after pembrolizumab initiation, he presented with dysarthria and disequilibrium. Upon neurological examination, the individual presented with left eye nystagmus, an ataxic gait, absent tendon responses in the lower limbs, vertigo, and mild dysmetria in the right upper and lower limbs. Brain MRI revealed micro-ischemic-type lesions devoid of clinical significance. A private physician initiated oral dexamethasone at a dosage of 8 mg/day, resulting in a minor improvement in symptoms. Subsequent cerebrospinal fluid (CSF) analysis disclosed elevated total proteins (112 mg/dL), increased glucose (87 mg/dL), and 20 cells per mm3, with lymphocyte predominance (92%) (Table 3). Moreover, the patient underwent a virological test and test for paraneoplastic antibodies, which both came back negative. Empirical treatment with ceftriaxone, acyclovir, and methylprednisolone at

a dosage of 2 mg/kg/day was initiated on the first day. After further tests, he was diagnosed with aseptic meningitis (Grade 4-CTCAE v5.0). Although the patient exhibited mild improvement during hospitalization, he did not experience complete symptom remission. Subsequent lumbar punctures and brain MRIs yielded consistent findings. Attempts to reduce methylprednisolone dosage resulted in symptom exacerbation. Following additional neurologic consultation, the patient commenced immunoglobulin G at a dosage of 2 g/kg for five infusions, in conjunction with methylprednisolone at a dosage of 1 mg/kg/day. While the patient experienced some improvement in his symptoms, complete remission was not achieved. Post discharge, the individual did not undergo further immunotherapy. In February 2021, second-line chemotherapy was administered due to disease progression. The patient succumbed in December 2021 due to complications arising from the disease.

### *3.3. Case 3*

A 60-year-old male was diagnosed with NSCLC with liver and brain metastases in September 2020. Following the completion of whole-brain radiation therapy in October 2020, the patient underwent chemotherapy combined with pembrolizumab (PD-L1 0%). By July 2021, due to response to the treatment, the patient was placed on pembrolizumab as maintenance monotherapy. During a visit to the outpatient clinic, 39 weeks after the initiation of immunotherapy, the patient reported experiencing muscle weakness in both lower limbs and headache. Subsequent diagnostic tests, including an electromyogram and a brain MRI, did not reveal any pathological findings to the contrary the brain MRI indicated a complete response to the previously diagnosed brain metastasis. Laboratory testing disclosed deficient cortisol levels, along with low adrenocorticotropic hormone (ACTH) and thyroid-stimulating hormone (TSH). A specialist endocrinology input was sought and a pituitary MRI was performed, revealing the pituitary gland's enlargement and a progressively enhancing bilateral ring-shaped enhancement (Table 3). Consequently, the patient was diagnosed with hypophysitis (Grade 3-CTCAE v5.0) and received hormone replacement therapy, resulting in full recovery of his symptoms. As of September 2023, the patient remains on maintenance therapy with pembrolizumab, with ongoing endocrinological monitoring.

### *3.4. Case 4*

A 68-year-old female patient was diagnosed with epithelioid pleural mesothelioma in 2009.

Following three lines of chemotherapy due to progressive disease, in August 2021, she initiated a combination of nivolumab plus ipilimumab. After 13 weeks of the initiation of immunotherapy, she presented with severe exhaustion, anorexia, dizziness, and nausea. Brain MRI revealed mild swelling of the pituitary gland, exhibiting characteristics suggestive of hypophysitis (Table 3). Laboratory testing indicated low serum cortisol, ACTH, and TSH levels. Consequently, an endocrine consultation was conducted, leading to the diagnosis of hypophysitis (Grade 3-CTCAE v5.0). Thus, the patient was placed on hydrocortisone replacement therapy and experienced full recovery. One month thereafter, she resumed nivolumab, a treatment that she continues to receive to date (09/2023) with a stable disease status. She is currently undergoing regular follow-ups with an endocrinologist and receiving hormone replacement therapy.

### 4. Discussion

In our retrospective analysis, we observed that immune checkpoint inhibitor (ICI)-related central nervous system (CNS) adverse events and hypophysitis were infrequent occurrences, representing 0.49% of our ICI-treated patients (4/812). Notably, no deaths were attributed to CNS AEs or hypophysitis related to ICI.

Within our clinical practice, we identified isolated instances of meningoencephalitis (0.12%), aseptic meningitis (0.12%), and hypophysitis (0.25%), with a median onset time of 17 weeks. All four cases necessitated inpatient care, with intravenous corticos-

teroids administered as the primary therapeutic intervention. Notably, three patients were corticosteroid-sensitive, experiencing full recovery, while one patient, resistant to corticosteroid treatment, additionally received immunoglobulin G. Due to the severity of symptoms, two patients abstained from further immunotherapy, while one continued monotherapy with nivolumab. Two individuals who experienced hypophysitis remain on immunotherapy following the initiation of hormone replacement therapy. As of the last data extraction, two patients are still alive.

Remarkably, immune-related encephalitis has been documented to occur in approximately 0.1–0.2% of patients receiving immune checkpoint inhibitors, a trend consistent with our findings [36–38]. Presently, it remains unknown which among the approved PD-1, PD-L1, or CTLA-4 inhibitors is primarily associated with causing encephalitis, given the scarcity of data related to the rarity of this adverse event [15,39,40]. Clinical manifestations commonly include psychiatric symptoms, altered mental status, memory loss, and seizures, which tend to be resistant to antiepileptic therapy, as previously noted. These symptoms may also coexist with meningitis, as observed in our patient. Magnetic resonance imaging (MRI) scans, particularly T2-weighted and FLAIR images, may reveal cerebral inflammation and epileptogenic activity, albeit nonspecifically. Electroencephalogram results may also indicate epileptogenic activity. Cerebrospinal fluid (CSF) findings may include oligoclonal bands, pleocytosis, and elevated protein levels, mirroring our case. Identification of specific CSF antibodies has been reported in cases of encephalitis following immunotherapy, with onconeural autoantibodies such as anti-Ma2, anti-Hu, anti-GAD, anti-Yo, anti-NMDA, anti-CASPR2, anti-LGI1, anti-GABABr, and anti-AMPAr detected in case series [41–45].

Moreover, aseptic meningitis was diagnosed in 0.12% of our patients, aligning with findings in the literature [46,47]. However, this specific neurological adverse event is typically associated with the use of anti-CTLA-4 immune checkpoint inhibitors, whereas our case involved the administration of an anti-PD-1 antibody [48,49]. Primary symptoms mimic those of infectious meningitis and include headache onset, fever, neck stiffness, and photophobia. Imaging techniques usually do not reveal abnormalities, with the diagnosis relying on clinical manifestations and compatible meningitis findings in CSF examination, devoid of infectious factors or carcinomatosis. When patients also experience changes in mental status, meningitis may be diagnosed as meningoencephalitis [43].

Hypophysitis linked to immune checkpoint toxicity is predominantly associated with the combined use of anti-CTLA4 and anti PD-1 therapies, reaching 10%, while its incidence is lower, at approximately 1–5%, with either monotherapy of the aforementioned inhibitors [15]. Clinical symptoms are atypical, with patients presenting with headache, fatigue, nausea, weight loss, temperature dysregulation, appetite loss, and mental state changes [45,50,51]. Pituitary hormones are affected, leading to deficits in TSH and ACTH that may result in central hypothyroidism and adrenal insufficiency. Altered levels of TSH and mainly T4 may be evident in laboratory tests, while hyponatremia may present if CRH is elevated. Identification of possible pituitary adrenal insufficiency is crucial, as it may be fatal if not treated promptly. Imaging techniques, particularly pituitary MRI, can detect gland enlargement, aiding diagnosis based on clinical and laboratory findings [33]. In our study, one patient received anti-CTLA-4 in combination with anti-PD-1 inhibitors, and another received only an anti-PD-1 inhibitor. In both cases, MRI revealed gland enlargement, and symptoms resolved upon initiation of hormone replacement, resulting in excellent survival.

The findings of our study underscore the rarity of immune-related CNS toxicities. While the overall percentage of patients affected by these adverse events is very low, our study suggests a slightly higher incidence than previously reported [15].

The advent of immunotherapy has revolutionized cancer management in both the oncology and hematology fields. The unique mechanism of action of immune checkpoint inhibitors significantly enhances patient survival and clinical benefits. Their ability to unleash T-cell activation by suppressing the tumors' negative impact on immune cells lies

behind their effectiveness, while at the same time, it seems to be their Achilles' heel, making ICI-related adverse events a pressing concern [36,37].

As fully described in the literature, the occurrence of irAEs is associated with better overall survival [52,53]; therefore, the prompt recognition of patients experiencing potentially fatal adverse events is crucial, as early treatment can be life-saving. Given the wide range of immune-related toxicities, a close collaboration between oncologists and various medical specialties is imperative. Neurology and endocrinology consultations were integral to our approach, and our patients were managed according to available guidelines with multidisciplinary support, resulting in the resolution of adverse events [15,54].

As the application of immunotherapy continues to expand across almost all tumor types and disease stages, the accompanying toxicities are anticipated to follow. Thus, reporting and documenting any rare side effects are essential, as they can be dangerous and potentially life-threatening. With an increasing number of patients exposed to immune checkpoint inhibitors, the underlying mechanisms of toxicity and autoimmunity are gradually being elucidated. In the era of newly applied targeted therapies and personalized medicine, clinicians and patients sometimes find themselves in uncharted waters. Hence, heightened awareness and close collaboration are essential for the prominent identification and management of immunotherapy-related adverse events.

**Author Contributions:** Conceptualization, A.L.A., G.Z. and D.M.; methodology, E.K., V.K. and N.T.; A.L.A., A.G. and M.Y.; data curation, A.L.A., A.G. and M.Y.; writing—original draft preparation, A.L.A., A.G., M.Y., G.Z., N.T., V.K., E.K. and D.M.; writing—review and editing, D.M.; supervision, D.M.; project administration. D.M. All authors have read and agreed to the published version of the manuscript.

**Funding:** This research received no external funding.

**Institutional Review Board Statement:** This study was conducted in accordance with the Declaration of Helsinki and approved by the Institutional Review Board of the General University Hospital of Ioannina (protocol code 1045/15-11-2023).

**Informed Consent Statement:** Written informed consent was obtained from living individual(s) for the publication of any potential data included in this article.

**Data Availability Statement:** The data presented in this study are available upon request from the corresponding author, after ethics review.

**Conflicts of Interest:** The authors declare no conflict of interest.

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
