# Peer review of "Exploring a Rarity: Incidence of and Therapeutic Approaches for Neurological Complications and Hypophysitis in Cancer Patients on Immune Checkpoint Inhibitors—A Single-Center Study"

_curroncol, doi:10.3390/curroncol30120766_

Round 1
Reviewer 1 Report
Comments and Suggestions for Authors
The study is worthwhile to provide a benchmark of CNS irAE. However, the manuscript needs some work in its presentation of methodology.
The authors fail to mention that they will not be looking for nor describing all grades of irAE. Did any occur besides grades 3-4? I assume no grade 5, as the authors do mention that no one died.
The authors also fail to describe how they categorize the grades and if there is any verification of that categorization. Is this according to CTCAE?
I look forward to the rewrite.
Comments on the Quality of English Language
The manuscript needs editing for grammar and flow.
Author Response
"Please see the attachment."

Reviewer 2 Report
Comments and Suggestions for Authors
This is an interesting report on CNS toxicity of ICIs in a single center. Of more than 800 pts treated with ICI, only 4 presented w/ G3-4 CNS-related irAEs. The authors describe the 4 cases in detail, including their management.
Even though this is a relevant topic, in my opinion, this work should be classified as a case series, rather than a "real world evidence". The small numbers prevent larger conclusions.
Please consider the following suggestions:
- Change the title to - A case series (instead of real world evidence) - this would reflect the manuscript better
- Why toxicities of < G3 were not included? I do not understand the rationale, especially when only 4 cases were encountered
- In line 31 > There are other ICIs beyond anti-PD(L)1 and anti-CLTA4. For example, anti-lag3 > the paragraph should be rephrased.
- Given only 4 cases have been presented, the authors should consider doing a literature review of case reports. I believe this could add to the manuscript.
- Although overall the manuscript can be well understood, it would benefit of an English review. There are some misplaced terms: e.g., "uneventful"- line 47
Comments on the Quality of English Language
- Although overall the manuscript can be well understood, it would benefit of an English review. There are some misplaced terms: e.g., "uneventful"- line 47
Author Response
"Please see the attachment."

Reviewer 3 Report
Comments and Suggestions for Authors
The authors describe 4 cases in this case series, however, have included 2 cases of hypophysitis in the 4 described cases. Hypophysitis is considered an endrocrine toxicity (Brahmer, JR et al. J Clin Oncol. 2018 Jun 10;36(17):1714-1768. doi: 10.1200/JCO.2017.77.6385. Epub 2018 Feb 14. PMID: 29442540; PMCID: PMC6481621.); and is managed as such with most patients being able to resume their immunotherapy subsequently. Thus, should not be categorized under "neurotoxicity". The two other cases (encephalitis & meningitis) are cases of neurotoxicity and the manuscript should be revised as a 2 case series. The authors should also make the distinction that resumption of IO was only undertaken in hypophysitis cases & both cases with neurotoxicity died with disease progression without resuming IO. THus the introduction, case description and discussion sessions need to be alterated to limit to neurotoxicity cases.
Comments on the Quality of English Language
- Line 37 "ICI blockade the shut down of T-cells" is unclear and needs to be revised
- Line 156 "...she fully recovered her neurologic symptoms while liver enzymes...". 'While' should be changed to 'and'
- Line 177: Do authors mean "maximum dose"? instead of minimum dose. Otherwise just state the dose (eg. 1 or 2 mg/kg etc)
Author Response
"Please see the attachment."

Reviewer 4 Report
Comments and Suggestions for Authors
The authors retrospectively analyzed nervous system irAEs in 812 solid tumor patients treated with ICI from 2018 to 2023. They found that 1) Although rare in frequency, 4 patients developed grade 3-4 irAEs of the central nervous system (CNS), 2) All patients were hospitalized and received steroids, 3 patients recovered, and the remaining 1 patient received additional IgG administration, and 3) All four patients recovered from the grade 3-4 irAE of the CNS. Based on these findings, the authors concluded that irAEs in the central nervous system are rare but potentially fatal AEs, and prompt diagnosis and treatment with systemic administration of steroid are required.
This MS is a valuable report on irAE of the CNS and is considered clinically important. References are also properly cited
Major comments
・It is not clear that this study has been approved by an ethics committee. Please describe this in the Material and Methods section.
・This MS focuses on irAE of the CNS and the early diagnosis is essential. Therefore, a table summarizing the symptoms, MRI findings, and CSF findings of the four patients will improve this MS and help readers’ understanding.
Author Response
"Please see the attachment."

Round 2
Reviewer 1 Report
Comments and Suggestions for Authors
Authors have addressed my concerns.
Reviewer 2 Report
Comments and Suggestions for Authors
I commend the authors for incorporating many of the suggestions provided by the reviewers. The article reads much better in comparison to the 1st version and is currently acceptable to be published in my opinion.
Reviewer 3 Report
Comments and Suggestions for Authors
The authors have adequately revised the manuscript.
Comments on the Quality of English Language
Minor tense errors that could be reviewed during proof reading